# Photocatalytic Degradation of Crystal Violet Dye under Visible Light by Fe-Doped TiO_2_ Prepared by Reverse-Micelle Sol–Gel Method

**DOI:** 10.3390/nano13020270

**Published:** 2023-01-08

**Authors:** Antonietta Mancuso, Nicola Blangetti, Olga Sacco, Francesca Stefania Freyria, Barbara Bonelli, Serena Esposito, Diana Sannino, Vincenzo Vaiano

**Affiliations:** 1Department of Industrial Engineering, University of Salerno, Via Giovanni Paolo II 132, 84084 Fisciano, Italy; 2Department of Applied Science and Technology and INSTM Unit of Torino Politecnico, Corso Duca Degli Abruzzi 24, 10129 Torino, Italy; 3Department of Chemistry and Biology “A. Zambelli” and INSTM Research Unit, University of Salerno, Via Giovanni Paolo II, 132, 84084 Fisciano, Italy; 4Interdepartmental Centre PolitoBIOMed Lab., Corso Duca Degli Abruzzi 24, 10129 Torino, Italy

**Keywords:** Fe-doped TiO_2_, TiO_2_ polymorphs, brookite, non-ionic surfactant, photocatalysis, crystal violet, visible light

## Abstract

A reverse-micelle sol–gel method was chosen for the preparation of Fe-doped TiO_2_ samples that were employed in the photodegradation of the crystal violet dye under visible light irradiation in a batch reactor. The dopant amount was varied to assess the optimal photocatalyst composition towards the target dye degradation. The photocatalysts were characterized through a multi-technique approach, envisaging XRPD and QPA as obtained by Rietveld refinement, FE-SEM analysis, DR UV−vis spectroscopy, N_2_ adsorption/desorption isotherms measurement at −196 °C, ζ-potential measurement, and XPS analysis. The physical-chemical characterization showed that the adopted synthesis method allows obtaining NPs with uniform shape and size and promotes the introduction of Fe into the titania matrix, finally affecting the relative amounts of the three occurring polymorphs of TiO_2_ (anatase, rutile and brookite). By increasing the Fe content, the band gap energy decreases from 3.13 eV (with undoped TiO_2_) to 2.65 eV (with both 2.5 and 3.5 wt.% nominal Fe contents). At higher Fe content, surface Fe oxo-hydroxide species occur, as shown by DR UV-vis and XP spectroscopies. All the Fe-doped TiO_2_ photocatalysts were active in the degradation and mineralization of the target dye, showing a TOC removal higher than the undoped sample. The photoactivity under visible light was ascribed both to the band-gap reduction (as confirmed by phenol photodegradation) and to dye sensitization of the photocatalyst surface (as confirmed by photocatalytic tests carried out using different visible-emission spectra LEDs). The main reactive species involved in the dye degradation were determined to be positive holes.

## 1. Introduction

In the last century, the rapid industrial growth worldwide has shown the impact of environmental pollution on public health [1,2]. In this problem, water pollution is a major environmental concern, especially concerning dye contamination of wastewater from the textile, printing, dyeing, leather, food, and cosmetic industries [3]. Generally, synthetic dyes are resistant to common wastewater processes, being released in the environment where they can persist for a long period, finally affecting organisms in the ecosystem [4]. Another drawback of the presence of dyes in wastewaters is that they can hamper sunlight penetration in contaminated water bodies. Different kinds of synthetic dyes (such as azo disperse, acidic, basic triphenylmethane etc.) exist, having different chemical structures; in particular, triphenylmethane dyes are commonly applied in various industrial applications. Among them, Crystal Violet (CV), commonly called gentian violet (IUPAC name: tris (4-(dimethylamino) phenyl)methylium chloride) is a cationic dye with one dimethylamino group on each phenyl ring, with relatively low cost and fair dyeing efficacy. It is estimated that about 10–20% of the total world production of CV is lost during synthesis and processing [5]. Because of its noxious effects on human health, CV is considered a hazardous chemical, and its use is prohibited in the aquaculture and food industries. Hence, the removal of CV from industrial wastewater is needed for the protection of terrestrial and aquatic ecosystems. Several studies have addressed the issues of biological, physical and chemical treatment of CV-containing effluents. Bio-treatments are, in general, ineffective due to CV’s recalcitrant nature [6]. Furthermore, microorganisms are unable to remove CV completely due to the presence of the dimethyl amino groups [7]. Physical (non-destructive) methods, such as adsorption on activated carbons, merely transfer the pollutant to another medium, which should be regenerated, thus causing possible secondary pollution [8]. The best results in terms of CV decolorization and mineralization were obtained by using chlorine or ozone treatments, but unfortunately such chemical treatments require high dosages of oxidizers, making them not cost-effective [9]. Advanced Oxidation Processes (AOPs) represent a possible alternative for water remediation; in particular, heterogeneous photocatalysis is the most attractive method for dye removal since it can be carried out under ambient conditions and may lead to total mineralization of organic carbon to CO_2_. Another advantage is the possibility of employing a photocatalyst, usually TiO_2_, that is generally easily available, non-toxic, chemically and mechanically stable, and has high turnover.

Undoped TiO_2_ is a semiconductor that can be excited by UV light (band gap in the 3.0–3.4 eV); this way, electrons (e^−^) are promoted from the valence band (VB) to the conduction band (CB), leaving as many positive holes (h^+^) in the VB [10]. In water, the photogenerated e^−^/h^+^ pairs interact with dissolved oxygen and/or water molecules producing several reactive oxygen species (ROS), i.e., superoxide anion radical (O_2•_^−^), hydrogen peroxide (H_2_O_2_), and hydroxyl radicals (•OH) [11] which, in turn, give rise to oxidation reactions responsible for pollutant (including organic dyes) degradation. The wide band gap of TiO_2_ hampers its practical environmental applications under solar light, since the semiconductor can only exploit the small UV fraction and not the visible fraction of the solar emission. One of the strategies to improve TiO_2_ absorption properties is reducing the band gap by TiO_2_ doping, for instance with transition metal ions with compatible ionic radius. Doping with Fe is one of the most common choices, as this metal is non-toxic, abundant in the earth and capable of red-shift TiO_2_ absorption [12,13,14,15,16]. Although doping is a key strategy to improve the optoelectronic properties of semiconducting metal oxides, its effectiveness and extent is greatly affected by the preparation method [17].

There is, indeed, considerable controversy over the effect of doping and the selection of the best dopant that can effectively extend the absorption spectrum in the visible light range and improve photocatalytic performance under the full irradiation of sunlight [18,19,20]. The fluctuating results reported in the literature can be mainly ascribed to the synthesis route, which can significantly impact the extent of doping, surface or bulk, the physicochemicalproperties and the optical response of the prepared material [19,21,22].

The literature reports different types of synthesis methods of undoped and doped TiO_2_, each of them characterized by pros and cons. [18,23,24,25,26,27]. The impregnation method is widely used in the preparation of catalysts due to the simplicity of experimental set-up, but it suffers from poor dispersion of the activity phase with rather limited substitution of the metal ions in the crystal lattice of TiO_2_, resulting in a poor photocatalytic performance [28].

Hydrothermal synthesis is a valuable alternative, although it is not free of drawbacks such as the need for expensive autoclaves and the use of temperatures above 120 °C for extended periods [29]. Furthermore, the use of a ‘black box’ makes monitoring the process impossible. Changing the synthesis parameters is therefore carried out without an accurate understanding of the step on which action is being taken [30].

In this scenario, the sol–gel method offers one-pot synthesis routes with high potential and a number of advantages over conventional synthesis methods, such as a low operating temperature and high homogeneity of the final product. Consequently, a large number of advanced materials whose properties are a function of the synthesis conditions have been prepared by the sol–gel method [31,32,33,34,35,36]. Among the different types of chemical processes that can benefit from the use of sol–gel chemistry, the reverse-micelle approach can be considered particularly effective in controlling the nucleation and growth of nanoparticles [37,38]. The peculiarity of the method lies in the possibility of confining reactions that usually take place in aqueous media to the small domains of reverse micelles. By this means, intimate contact between the precursors is encouraged, promoting homogeneity and mixing on an atomic scale and ultimately promoting the inclusion of the dopant in the oxide lattice.

Several papers report the photocatalytic degradation of CV dye under UV light in the presence of TiO_2_- and ZnO-based photocatalysts [39,40,41,42,43]. However, despite the CV dye degradation under visible light also being studied using different photocatalyst formulations (such as BiO_x_Cl_y_/BiO_m_I_n_ composites [44], BiVO_4_/FeVO_4_ [45], CdS-anchored porous WS_2_ [46] and β-Cu_2_V_2_O_7_) [47], the use of Fe-doped TiO_2_ for the removal of this dye is still poorly investigated [48]. Moreover, no paper about the doping of TiO_2_ lattice with iron by the reverse-micelle template-assisted sol–gel method for the degradation of CV dye is available in the literature. For these reasons, the aim of this work is to investigate the photocatalytic performances of Fe-doped TiO_2_ samples prepared using such a method for the degradation of CV dye under visible light.

## 2. Materials and Methods

All the reagents for the photocatalysts synthesis were bought from Merck-Sigma Aldrich Europe (Schnelldorf Distribution). Undoped and doped TiO_2_ photocatalysts were synthesized by a reverse-micelle (RM) sol–gel method as detailed in the following: the di-block copolymer (polyoxyethylene (20) oleyl ether, also known commercially as Brij O20) was dissolved in a cyclohexane (oil) phase by stirring at 50 °C. Then, MilliQwater was used to dissolve proper amounts of FeCl_3_ 6H_2_O (corresponding to the nominal contents of 1.0, 2.5 and 3.5 wt% Fe); the resulting solution was mixed into the cyclohexane mixture and stirred for 45 min, forming a water-in-oil (w/o) microemulsion in which micelles act as surfactant-encapsulated aqueous nanoreactors. Titanium(IV) butoxide (Ti(O(CH_2_)_3_CH_3_)_4,_ 98%) was then slowly added to the microemulsion, which was isothermally (50 °C) stirred for 2 h. Finally, the microemulsion was broken through the addition of 2-propanol, and sonicated. The resulting solid was separated by centrifugation, dried at 100 °C for 12 h, and calcined for 2 h in air at 500 °C with a temperature increase rate of 2.5 °C min^−1^ to burn the surfactant and promote crystallization [13,49]. The resulting samples will be hereafter referred to as RM_0Fe (undoped TiO_2_) and RM_xFe, where x = 1, 2.5 and 3.5 stand for the three wt% Fe studied.

Powder X-ray diffraction (XRD) patterns of Cu Kα radiation were obtained by a X’Pert nPhilips PW3040 (Panalytical, Almelo, Netherland) diffractometer in 2θ range = 10–100°; with a step = 0.026° 2θ and a time per step = 0.8 s. The diffraction peaks were indexed by referring to the Powder Data File database (PDF 2000, International Centre of Diffraction Data, Pennsylvania). The X’Pert High Score Plus 3.0e (Malvern Panalytical Ltd., Malvern, UK) software allowed calculation of the crystallites’ average size (D), which was determined by the Williamson–Hall plot. Quantitative Phase Analysis (QPA) was carried out on the X’Pert High Score Plus 3.0e software to evaluate the different phase contents by applying the full-profile Rietveld method to the XRD patterns.

FE-SEM (Field Emission Scanning Electron Microscopy) micrographs of the powders were taken on a Merlin FESEM instrument (Carl-Zeiss AG, Oberkochen, Germany).

The powders Specific Surface Area (SSA) was obtained by means of the measurement of N_2_ adsorption/desorption isotherms at −196 °C (Quantachrome Autosorb 1C, Boyton Beach, FL, USA) on powder samples previously outgassed at 150 °C for 4 h to remove water and other atmospheric contaminants. The SSA values were calculated according to the BET (Brunauer–Emmett–Teller) method. X-Ray Photoelectron Spectroscopy (XPS) has been performed on a PHI 5000 VersaProbe (ULVAC-PHI, Physical Electronics Inc., Kanagawa, Japan) instrument, equipped with monochromatic Al Kα radiation (1486.6 eV energy) as X-ray source. Two different pass energy values were used for the survey (187.75 eV) and HR spectra (23.5 eV). During the measurements, the charge compensation was obtained with a combination of an electron beam and low-energy Ar beam system. The HR spectra were curve-fitted by using the Casa XPS software (Casa Software Ltd.).

Diffuse Reflectance (DR) UV–vis spectra were obtained on powder samples by a Cary 5000 UV–vis-NIR spectrophotometer (Varian Instruments, Mulgrave, Australia) equipped with a DR apparatus.

ζ-potential profiles of the TiO_2_ and doped samples were obtained by evaluating the electrophoretic mobility as a function of pH. Electrophoretic light scattering (ELS) on a Zetasizer Nano- ZS (Malvern Instruments, Worcestershire, UK) was used. In a typical measurement, a powder suspension in ultrapure water (MilliQ), is magnetically stirred for 5 min. The ζ-potential was measured at room temperature by gradually varying the pH by with 0.1 M NaOH or 0.1 M HCl solutions.

The photocatalytic tests were carried out using 100 mL CV aqueous solution (initial concentration = 10 ppm) in a cylindrical pyrex photoreactor (ID = 2.6 cm, L_TOT_ = 41 cm and V_TOT_ = 200 mL) operating in batch mode. White, Green or Blue LED strips (emission range: 400–800 nm; irradiance: 16 Wm^−2^) were positioned one at a time around the external body of the photoreactor to irradiate the overall volume of the CV solution. Next, 3g L^−1^ of powder was added to the CV solution and the suspension was continuously mixed using a magnetic stirrer. Adsorption/desorption equilibrium of the dye on the photocatalyst surface was first achieved in dark conditions after 60 min and then the photocatalytic test was started under visible light irradiation (total duration: 180 min). During the entire duration of each test, an air flow (Q_air_ = 150 cm^3^min^−1^ (STP)) was fed into the reactor, through an air distributor device, to avoid dissolved oxygen limitation in the reaction medium. A fan cooling system maintained the system temperature in the range of 25–30 °C. During the tests, ca. 3 mL suspensions were withdrawn from the photoreactor at regular times, centrifuged and analyzed by a UV-Vis spectrophotometer (Thermo Scientific Evolution 201, Thermofisher Italia, Monza) to assess the photoreaction progress. More in details, the CV decolorization was monitored by evaluating the CV solution absorbance at 583 nm [50]. In addition, the total organic carbon (TOC) of the treated solutions was detected. The TOC was evaluated from CO_2_ evolved during the high-temperature (680 °C) catalytic oxidation of liquid samples. CO_2_ produced in the gas phase was monitored by continuous analyzer ((Uras 14, ABB spa, Sesto San Giovanni, Milan, Italy)) [51].

## 3. Results and Discussion

### 3.1. Samples Characterization

Figure 1 reports the XRD patterns of the four samples, showing the occurrence of the peaks of the three most common polymorphs of TiO_2_, namely anatase (JCPDS file: 01-078-2486), rutile (JCPDS file: 01-083-2242) and brookite (JCPDS file: 96-900-4138). In the undoped sample, the rutile phase is less abundant, as evidenced by the presence of only one, very weak, peak at 27.4 2θ, which becomes more intense at higher Fe content, when other peaks of rutile are also observed. Figure 1 shows that several peaks due to different TiO_2_ polymorphs may overlap, especially here, where broad peaks are observed, due to small crystallite size (vide infra).

Quantitative Phase Analysis was carried out by Rietveld refinement to quantitatively evaluate the amount of the different occurring phases; the corresponding results are reported in Table 1, along with the crystallite size of the different polymorphs of TiO_2_.

As mentioned above, the rutile content was negligible in RM_0Fe (Table 1). The addition of nominal 1 wt.% Fe (sample RM_1Fe) leads to an increase in the rutile content, likely at the expenses of anatase, whereas the relative abundance of brookite does not change considerably compared to RM_0Fe, but at higher Fe content (samples RM_2.5Fe and RM_3.5Fe), the relative abundance of brookite considerably increases (Table 1).

The QPA results are confirmed by the increased intensity of the peak at 30.8 2θ, ascribable only to brookite and by both the decreased intensity of other peaks assignable only or manly to anatase (for instance the peaks at 53.8, 55.0 and 62.6 2θ) and the increased half-width of some peaks, to which both anatase and brookite reflections may contribute. According to the QPA results, by increasing the nominal Fe content, the amount of anatase decreases and, correspondingly, the amounts of both brookite and rutile increase (samples RM_2.5Fe and RM_3.5Fe). This effect may be due to the presence of Fe^3+^ ions within the micelles’ core, which notoriously give rise to a more acidic environment, a condition favouring the formation of brookite [52], as already observed to occur by using the same synthesis procedure with other metallic dopants [37,38,53]. The formation of rutile, notwithstanding the adopted calcination temperature, can be due to the presence of both brookite, which may favour the anatase-to-rutile thermal transition [54,55], and Fe^3+^ ions; indeed, as reported by Hanaor et al. [56], low charge cations (i.e., <+4) can act as anatase-to-rutile transition promoters.

Due to the overlapping of the main reference peaks of the three polymorphic phases and the very small dimension of the crystallites, the sizes reported in Table 1 have to be considered with care, but provide an indication of the Fe doping effect on the final material. Figure 2a reports the variation in size of the three polymorph crystallites with the nominal Fe content: the effect of Fe doping on the size of brookite’s crystallites is limited, whereas it is more pronounced with both rutile and anatase crystallites. The same considerations about peak overlap and width hamper a thorough analysis of how the cell parameters vary with the Fe content, although if the cell volumes are considered for the three polymorphs, the largest change is observed with those of anatase and brookite at 1 wt.% Fe (Figure 2b). In a previous work concerning doped TiO_2_ powders [38] obtained by the same reverse-micelle sol–gel method, an increase in the anatase cell volume with low dopant content followed by an increase in the cell volume at higher dopant contents have been ascribed to the initial formation of a substitutional solid solution forms, leading to an expansion of the unit cell, followed by a cell volume decrease at higher dopant concentration, as extra dopant ions cannot enter the TiO_2_ lattice anymore, being therefore forced into interstitial sites. Here, we observed this trend also with the brookite cell volume; since the content of rutile is hardly measurable in the RM_0Fe, no further considerations can be made for the cell volume of this polymorph. As a whole, XRD analysis shows that Fe doping modifies the cell volumes of both anatase and brookite, confirming the synthesis method’s effectiveness.

Figure 3 reports selected FE-SEM micrographs of the undoped TiO_2_ sample (RM_0Fe) and the RM_1Fe and RM_2.5Fe powders: the three powders show round shape NPs, with uniform shape and size (ca. of the crystallite size), independently of the Fe content, forming agglomerates/aggregates, in agreement with previous works concerning Mo- and Mn-doped TiO_2_ NPs obtained by the reverse-micelle sol–gel method [13,37,38], which allows controlling the growth of NPs within the micelles, as described in the Introduction.

In Figure 4, the N_2_ isotherms measured at −196 °C on the studied powders are reported: the samples show a type IV isotherm with an H2-type hysteresis loop, due to multi-layer adsorption and capillary condensation within inter-particle mesopores, formed upon calcination, a procedure leading to the formation of dense NPs organized into the porous agglomerates/aggregates observed by FESEM (Figure 3).

The corresponding values of BET SSA and total pore volume (Table 2) show that doping with 1 wt.% Fe has a limited effect on the surface properties, whereas 2.5 wt.% Fe leads to both higher porosity and SSA, whereas 3.5 wt.% Fe leads to a decrease in SSA. Such a behaviour observed at the highest Fe content could be due to the formation of some Fe-containing surface species, which may escape XRD detection due to their small size and overall Fe amount [40]. Accordingly, the XPS-determined surface Fe/Ti atomic ratio in the RM_3.5Fe sample (equal to 0.120, i.e., exceeding the nominal Fe/Ti atomic ratio of 0.052, Table 2) may be due to an overall agglomeration/aggregation of Fe-containing species at the samples surface.

**Figure 4 nanomaterials-13-00270-f004:**
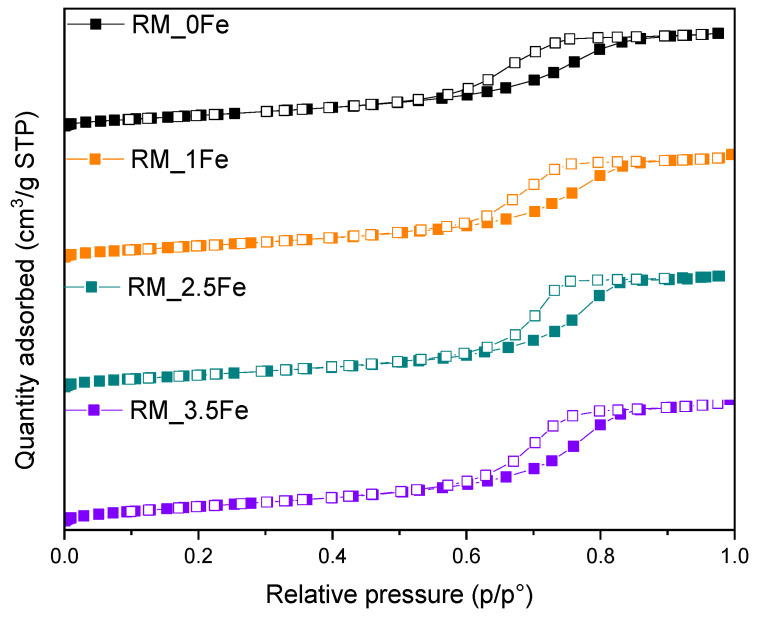
N_2_ adsorption/desorption isotherms at −196 °C on the studied samples: RM_0Fe (black squares), RM_1Fe (orange squares), RM_2.5Fe (green squares) and RM_3.5Fe (violet squares). Full and empty symbols refer to adsorption and desorption run, respectively.

**Table 2 nanomaterials-13-00270-t002:** Nominal Fe content (wt.%), nominal Fe/Ti atomic ratio and surface Fe/Ti atomic ratio (as determined by the survey XP Spectra in Figure 5); BET SSA and total pore volume, as obtained by N_2_ isotherms at −196 °C; band gap energy (Eg, eV) as obtained by linear extrapolation of the absorption edge (a) and by applying the Tauc’s plot method for indirect band gap semiconductors (b, Figure 5); pH_IEP_ values as determined by electrophoretic measurements.

Sample	Nominal Fe Content (as Fe wt.%)	XPS Determined Surface Fe/Ti Atomic Ratio	BET SSA (m^2^ g^−1^)	Total Pore Volume (cm^3^g^−1^)	Band Gap Energy (E_g_, eV)	pH_IEP_
	Nominal Fe/Ti Atomic Ratio					
RM_0Fe	0	0	105.7	0.203	3.31 ^a^	3.5
					3.13 ^b^	
RM_1Fe	1.0	<0.0048	106.4	0.212	3.22 ^a^	3.6
	0.0414				2.97 ^b^	
RM_2.5Fe	2.5	0.021	116.1	0.249	3.00 ^a^	3.8
	0.037				2.65 ^b^	
RM_3.5Fe	3.5	0.120	98.3	0.252	3.03 ^a^	3.8
	0.052				2.65 ^b^	

**Figure 5 nanomaterials-13-00270-f005:**
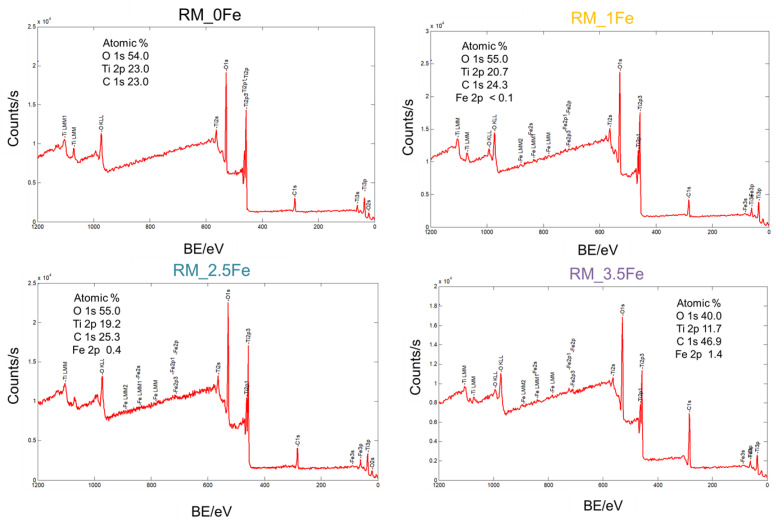
Survey XP spectra from which the surface Fe/Ti atomic ratio values in Table 2 have been determined. With all the samples, besides O, Ti (and Fe in doped ones) no other elements besides normally occurring adventitious C were detected.

Figure 6a reports the DR UV-Vis spectra of the studied powders; the undoped TiO_2_ sample shows an intense absorption with an onset at about 410 nm, due to charge transfer transition from O^2-^ to Ti^4+^ species. Fe is a dopant normally adopted to shift towards the Vis range the absorption onset of TiO_2_ [54]. As expected, Fe doping induces a (bathochromic) shift of the onset of adsorption of the studied samples, but also to the formation of a broad absorption at ca. 490 nm (asterisk), likely due to the d-d transitions of Fe^3+^ ions [57,58,59] in Fe-oxohydroxo clusters at the surface of the NPs, as aleady observed by some of us in other types of systems [60] and in other Fe-doped TiO_2_ powders [53,61].

Interestingly, with the RM_1Fe sample, a strong absorption band at ca. 230 nm is observed, likely due to to charge transfer transition from O^2-^ to octahedral Ti^4+^ ions strongly perturbed by the introduction of Fe^3+^ ions in the bulk of TiO_2_, as previously reported [62], and in agreement with the previously reported XRD data. The signal due to Fe-oxohydroxo clusters is more evident with higher Fe contents (with sample RM_2.5Fe and, especially, sample RM_3.5Fe) and has a limited intensity; however, determining the relative amount of isomorphically substituted Fe^3+^ with respect to the amount of Fe occurring in the clusters is not a trivial issue, due to the poor quantitative character of DR UV–Vis spectroscopy. The band gap energy (E_g_) of the materials has been determined both by the Tauc’s plot method (Figure 6b) and by extrapolation of the absorption onset of the spectra in Figure 6a (both values are reported in Table 2), since the Tauc’s plot method makes an assumption about the type of electronic transition of the studied semiconductor. Here, since anatase is the most abundant phase, we considered Tauc’s method around an indirect semiconductor, but since the brookite (and rutile) content increases with Fe doping, we also reported the value extrapolated from the spectra (both being brookite and rutile direct semiconductors). As a whole, increased Fe-doping leads to a decrease in the E_g_ values, up to 2.5 wt.% Fe, after which a higher Fe content does not lead to a significant change in the E_g_, possibly due to the formation of larger Fe-oxohydroxo clusters on the surface of TiO_2_.

Figure 7 reports the HR XP spectra concerning the Ti 2p line (Figure 7a) and the O 1s line (Figure 7b) of the studied samples. First of all, as normally occurs with doped samples, i.e., when the amount of the doping element is low with respect to the parent matrix, the Fe 2p lines (insets of Figure 7a) were weak in intensity and extremely noisy, finally hampering any kind of curve-fitting procedure, since in order to study Fe^2+^/Fe^3+^ oxidation states, reference should be made to satellite peaks, which are notoriously weaker than the corresponding lines [63].

Analysis of the Ti 2p line (Figure 7a) is, however, very useful: the Ti 2p spectral region showed the typical spin-orbit splitting doublet of Ti 2p_3/2_ and Ti 2p_1/2_ states, the observed splitting being constant and equal to 5.7 with all the studied samples, confirming the sole presence of surface Ti^4+^ species [64]. Interestingly, the Ti 2p peaks were observed to shift to lower BE values in doped samples, an effect already observed with Fe-doped TiO_2_ [65] and assigned, in the literature, to Fe species affecting the electronic state of Ti, probably as a consequence of Ti ion substitution by Fe^3+^ ions in the lattice. Such a result further corroborates the previously discussed XRD and DR UV-vis results, especially with the RM_1Fe sample, showing the larger variation in the anatase and brookite cell volume and a band at 230 nm, assigned to Ti^4+^ perturbed by nearby Fe^3+^ ions.

Concerning the O 1s line (Figure 7b), with all the samples, two components were observed: according to the literature, the one at the lower BE is assigned to lattice oxygen (O^2-^) species, and that at the higher BE to surface adsorbed water/OH groups [65]. Interestingly, the relative abundance of the higher BE component is much higher with the RM_3.5Fe sample, and could be due to the progressive formation of Fe-oxohydroxo clusters.

Figure 8 shows the ζ-potential curves of the studied samples: surface Ti-OH groups have an amphoteric behaviour, being protonated and deprotonated below and above the pH_IEP_ (pH at the IsoElectric Point), respectively, affecting the possible repulsion/attraction occurring with the CV dye, which is a cationic species at the natural solution pH. With a set of undoped TiO_2_ samples obtained by different synthesis methods [66], we have found different values of pH_IEP_; with TiO_2_ nanoparticles, various pH_IEP_ values are indeed reported in the literature and very different justifications are proposed, such as particle size, synthesis method, type of polymorph, etc. [67]. Therefore, the results reported here are measured in the same experimental conditions allowing a comparison between samples to assess (i) the possible effect of Fe doping and (ii) the kind of interaction possibly taking place with the dye moieties.

The pH_IEP_ of the undoped sample RM_0Fe (equal to 3.5, Table 2) is very close to the value of 3.56 measured for another batch obtained by the same synthesis method [62]. Doping with Fe leads to slightly higher pH_IEP_ values, but does not bring about a marked change in the pH_IEP_ value, which increases slightly with RM_1Fe and RM_2.5Fe, indicating that the effect of the surface Fe species is limited, as pH_IEP_ values of Fe_2_O_3_ [67] are found at higher pH (between 5.80 and 6.20). The dashed lines at ζ = + 30 mV and −30 mV show the range of ζ potential in which, according to the literature, NPs suspensions are stable: all the studied NPs are positively charged at very low pH values, and their low surface change (in absolute value) in the whole pH range could affect the suspension stability as well as any adsorption/desorption phenomena.

Summarizing the physicochemical characterization of the studied powders discussed so far, Fe-doping has an effect on both their structural and surface properties, affecting the overall phase composition, favouring the formation of brookite, lowers the E_g_ value, and induces the formation of surface defects, likely Fe-oxohydroxo clusters. Moreover, considering the set of samples under study, the RM_2.5Fe sample is characterized by the highest SSA and lower band gap value and is promising from the point of view of photocatalytic applications under visible light.

### 3.2. Photocatalytic Activity Tests

The degradation of the CV dye under visible (white) light irradiation in the presence of the undoped photocatalyst RM_0Fe and of the RM_xFe-doped TiO_2_ is shown in Figure 9. Dark adsorption, also displayed in Figure 9, evidenced a decreasing adsorption of the dye with the increase in the nominal Fe content. This behavior is probably related to the decrease of anatase content, which seems to be the most favorable polymorph to the CV-surface interaction, since no significant differences in the pH_IEP_ values have been observed (Figure 8) between the undoped and doped TiO_2_. Even if strong differences were observed in CV adsorption capacity, since the undoped sample is the most efficient at the dark removal of CV, all the prepared photocatalysts, except for RM_3.5Fe, reached high values of CV discoloration. In details, RM_0Fe, evidenced a CV discoloration of about 96% within 60 min of irradiation, RM_1Fe and RM_2.5Fe led to a similar CV discoloration after 180 min of irradiation, while at a higher Fe amount (RM_3.5Fe), the CV discoloration was lower and equal to 75%. The high value of discoloration under visible irradiation with the undoped TiO_2_ (RM_0Fe) could be attributed to the possible sensitization by the CV dye. Nevertheless, Table 3 reports the samples mineralization efficiency as TOC removal % after 180 min under visible (white) irradiation. The mineralization efficiency for RM_0Fe was largely negligible, and this must be remarked upon to understand the selection of the best photocatalyst; indeed, the TOC removal with RM_2.5Fe photocatalyst was the highest, with a mineralization efficiency equal to about 52%, while lower values of TOC removal were found with the other samples. The higher TOC removal of the Fe doped samples with respect to the undoped TiO_2_ could be related to the presence of Fe^3+^ ions in the crystalline TiO_2_ structure that allow enhancing the separation of the photo-generated charge carriers, since Fe^3+^ can act as electron acceptor, consequently reducing recombination phenomena of e^−^/h^+^ pairs [53]. Conversely, at higher Fe-contents, the formation of surface Fe-containing species can lead to the undesired formation of recombination centers, finally lowering the photocatalytic yield.

In order to distinguish the influence of different wavelength ranges of irradiation on the photocatalytic discoloration of CV, the performances of white, green and blue LED were compared. The trends of CV relative concentrations under different irradiation sources are shown in Figure 10; the results show that all the studied irradiation sources were able to decolorize CV. However, the best performances were achieved with the white LED, in that 180 min of irradiation allowed achieving 96% CV decolorization as well as a TOC removal of 52%.

The green LED, emitting a light peak centered at 518 nm (Figure 11a), can only excite the CV moiety, which presents a large absorption maximum at 583 nm (Figure 11b). With the green LED, the color removal is lower than with the white LED since there is a lack of stronger energetic radiation. Surprisingly, even with the (more energetic) radiation of the blue LED (emission peak at 466 nm) there is no increase in decolorization, and the related curves appear similar to each other.

This trend suggests that two contributions are responsible for the observed increased color removal with white LED, the semiconductor excitation and the photosensitization with the dye. Indeed, the band gap value of the RM_2.5Fe sample is 2.65 eV (Table 2); the Planck law [48], λ = hc/E_g_, allows calculating an onset of excitation wavelength of 468 nm. In the white light two main emission peaks occur at 440 nm (sharp) and around 600 nm (broad). Therefore, while the blue LED can just activate the doped semiconductor, the more intense irradiation centered at 440 nm of the white LED, emitting comparable amounts of more energetic photons, corresponds to more photogenerated charge carriers.

Furthermore, in order to investigate the role of bubbled gas inside the suspension on CV decolorization, nitrogen and air were fed into the reaction solution. The experimental results (Figure 12) evidenced that, when air is continuously bubbled, both CV decolorization and mineralization are higher than when bubbling nitrogen.

This result was probably due not only to •OH radicals, produced to degrade CV, but also to other species, such as superoxide radical anions (O_2_•^−^) generated in the presence of air that participate in the chemical reactions [68].

TOC removal results are reported in Table 3. It must be noticed that green and white LEDs induce a TOC removal around 50%; meanwhile, the blue LEDs are able to account for only 65% of white LED TOC removal, indicating that the blue LEDs are less intense, being at the emission at the onset of catalyst absorption. The result in the presence of nitrogen is interesting as well, in that it shows a lower TOC removal with respect to the air bubbling, suggesting a non-negligible role of superoxide radical anion.

However, in order to deepen the understanding of CV photocatalytic degradation mechanism with the RM_2.5Fe photocatalyst, different scavengers were added to trap active species, identifying the species responsible for CV degradation in such a way by the decrease in photoactivity (Figure 13); p-Benzoquinone (BQ), ethylenediaminetetraacetic acid (EDTA) and isopropyl alcohol (IPA) were used as a scavengers in this study to quench h^+^ (holes), O_2_•^−^ and •OH respectively. With the addition of these three scavengers at a concentration equal to 10 mmol L^−1^, the discoloration rate was reduced. EDTA (holes scavenger) had a major effect on photodegradation and, as a consequence, the holes are the primary reactive species in the dye degradation process. With BQ (superoxide radical scavenger) and IPA (hydroxyl radical scavenger) minor effects were observed, indicating that the superoxide and hydroxyl radicals are not the primary reactive species in the dye decolorization process.

The reusability of the photocatalyst RM_2.5Fe with regard to the CV discoloration is shown in Figure 14, where up to four cycles were performed on the same sample of photocatalyst after recovery and washing with distilled water without any further thermal treatment. The comparison gives evidence that after the first cycle, the values of CV color removal after 180 min were almost the same, showing promising stability.

Finally, with the aim to verify the photocatalytic efficiency on a colorless model pollutant, a photocatalytic test with a phenol solution (10 mg L^−1^) was performed dosing 0.3 g of the optimized RM_2.5Fe photocatalyst in 100mL under visible (white) light irradiation. Figure 15 shows about 37% phenol photodegradation efficiency, allowing us to affirm that the optimized RM_2.5Fe photocatalyst is able to effectively degrade both CV dye and also very stable organic compounds, such as phenol, which is widely recognized in wastewater and associated with environmental hazards and concern over public health [50]. Furthermore, in the case of phenol, no sensitization of the photocatalyst surface occurred, demonstrating the actual activation of RM_2.5Fe by visible light. Moreover, the level of phenol removal after 180 min was comparable to that observed in other works [69,70] or even higher [71].

## 4. Conclusions

Undoped and Fe-doped photocatalysts active under visible light were successfully obtained by reverse-micelle synthesis, a method that allows obtaining nanostructured TiO_2_ and introducing heteroatoms (here, Fe) within the TiO_2_ matrix. The undoped TiO_2_, efficient towards CV discoloration (likely due to sensitization effects) was, instead, inefficient in the mineralization of the CV. Doping with Fe leads, instead, to the relevant TOC removal of CV from aqueous solutions within 180 min under visible irradiation, the best (doped) catalyst having an optimal nominal Fe content of 2.5 wt.%. Likely, higher Fe content mainly leads to the formation of surface Fe-oxohydroxo clusters (as detected by DR UV-Vis spectroscopy) and to agglomeration/aggregation phenomena of Fe (as identified by XPS chemical analysis) at the surface of TiO_2_, where such Fe-containing species can finally act as charge-recombination centers that lower the photocatalytic activity; when such species were present, a decrease in the mineralization of CV was, indeed, observed. The presence of Fe also affected the type of TiO_2_ polymorphs occurring in the studied powders, in that the overall amount of anatase decreased, whereas that of brookite (and, to a minor extent, of rutile) increased, a fact that could also prevent electron/hole recombination.

The experiment with scavengers evidenced the major role of positive holes on the photocatalytic activity. Moreover, the 2.5 wt.% Fe-containing sample was also able to degrade phenol, a very harmful species, under visible light, likely due to its efficient exploitation of the visible light in virtue of its band gap energy, since any sensitization could occur with such contaminant.

## Figures and Tables

**Figure 1 nanomaterials-13-00270-f001:**
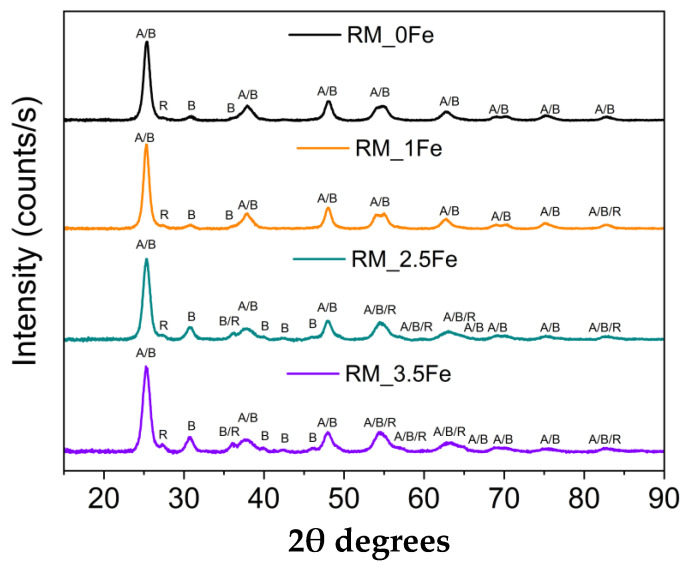
Powder XRD patterns in the 10–90 2θ degrees range of the following samples: RM_0Fe (black curve), RM_1Fe (orange curve), RM_2.5Fe (green curve) and RM_3.5Fe (violet curve). The A, B and R labels identify the main peaks of anatase, brookite and rutile, respectively.

**Figure 2 nanomaterials-13-00270-f002:**
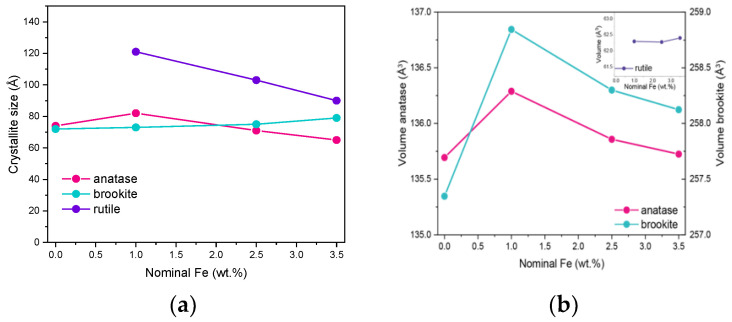
(**a**) Trend of the crystallite size versus the nominal Fe content (wt.%) for anatase, rutile and brookite (s) and trend of the cell volumes of the three polymorphs versus the nominal Fe content (wt.%) (**b**).

**Figure 3 nanomaterials-13-00270-f003:**
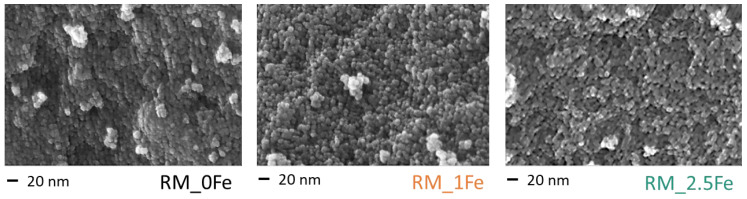
Selected FE-SEM micrographs as obtained with the RM_0Fe, RM_1Fe and the RM_2.5Fe powders, all showing similar morphology and degrees of agglomeration/aggregation.

**Figure 6 nanomaterials-13-00270-f006:**
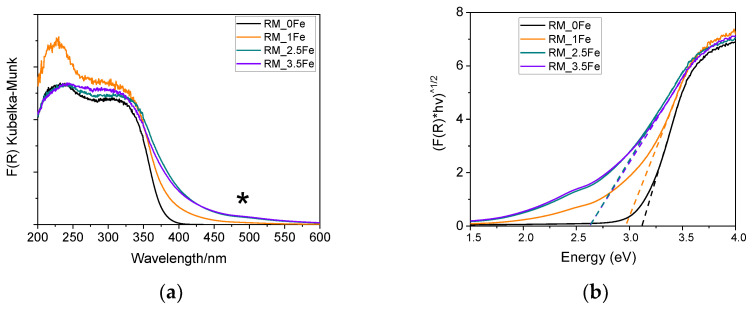
(**a**) DR UV-Vis spectra of the following powder samples at room temperature: RM_0Fe (black curve), RM_1Fe (orange curve), RM_2.5Fe (green curve) and RM_3.5Fe (violet curve). The asterisk at ca 490 nm indicates the possible d-d transitions of Fe3+ ions. (**b**) corresponding Tauc’s plots, as obtained by assuming indirect semi-conductor behavior.

**Figure 7 nanomaterials-13-00270-f007:**
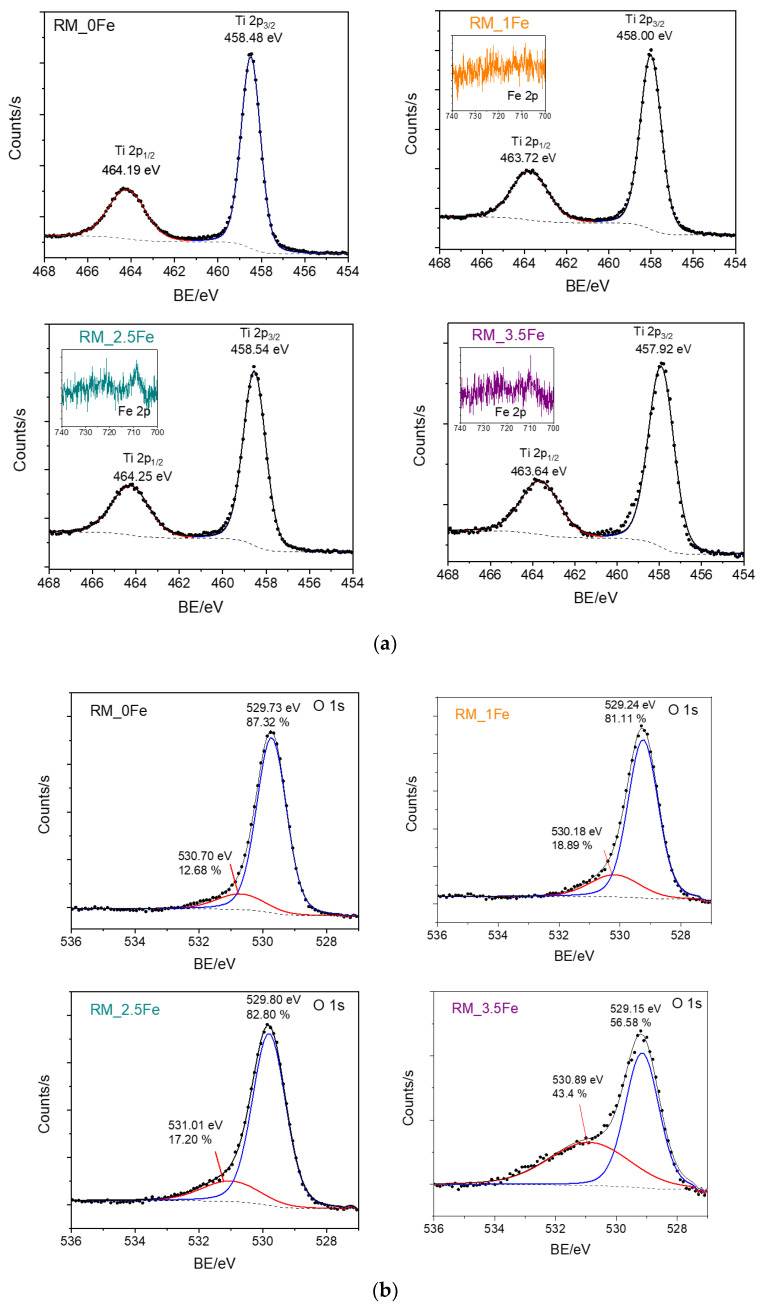
HR XP spectra concerning the Ti 2p line (**a**) and the O 1s line (**b**) of the studied powder samples. Insets of Section show a concern the Fe 2p line measured with Fe-doped samples, namely RM_1Fe, RM_2.5Fe and RM_3.5Fe.

**Figure 8 nanomaterials-13-00270-f008:**
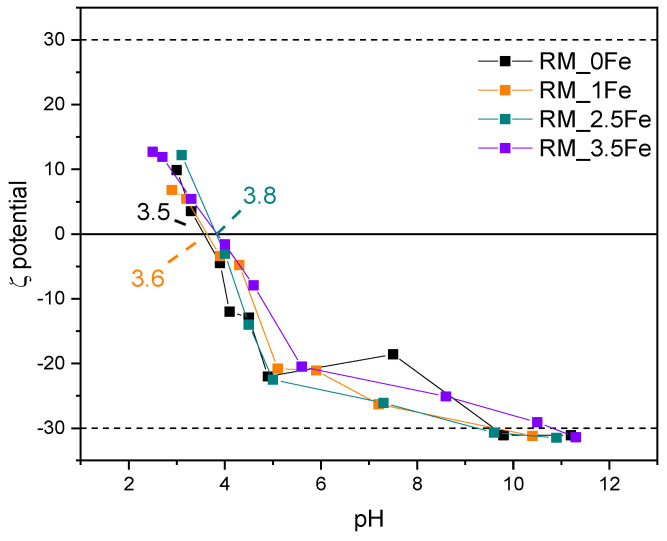
ζ-potential curves of the studied samples: RM_0Fe (black curve), RM_1Fe (orange curve), RM_2.5Fe (green curve) and RM_3.5Fe (violet curve).

**Figure 9 nanomaterials-13-00270-f009:**
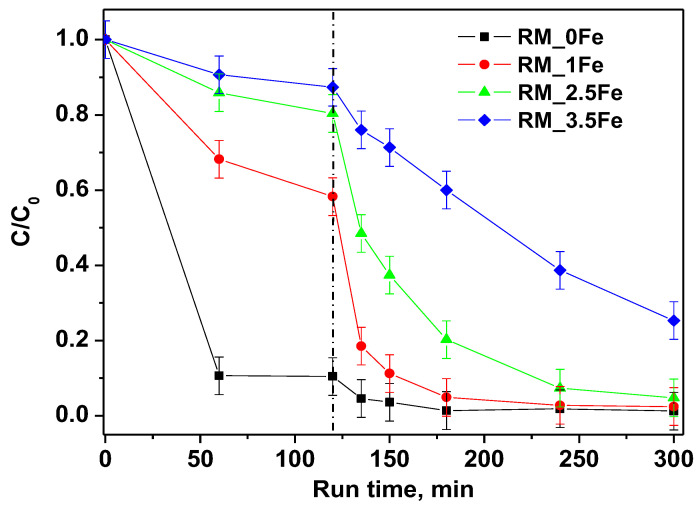
Photocatalytic discoloration of CV aqueous solution with all the studied photocatalysts under visible irradiation (white light).

**Figure 10 nanomaterials-13-00270-f010:**
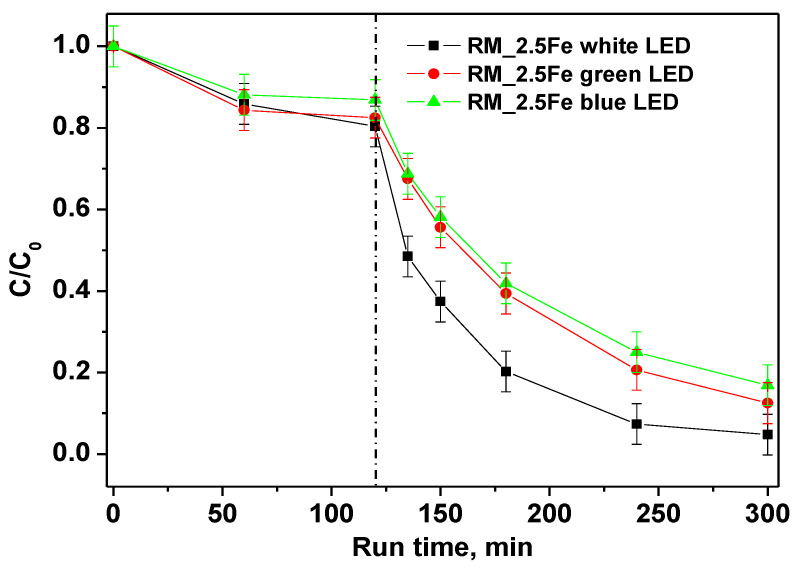
Photocatalytic discoloration of CV aqueous solution with the RM_2.5Fe photocatalyst under white, green and blue light irradiation.

**Figure 11 nanomaterials-13-00270-f011:**
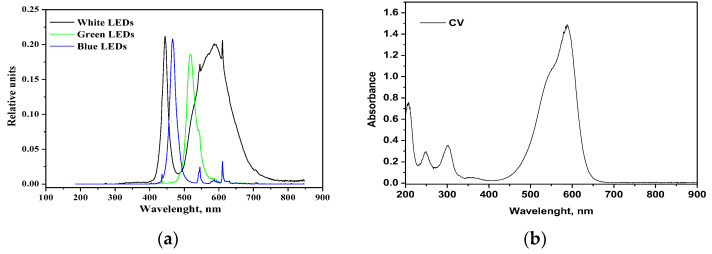
(**a**) Emission spectra of white, green and blue LEDs; (**b**) absorption spectrum of CV aqueous solution.

**Figure 12 nanomaterials-13-00270-f012:**
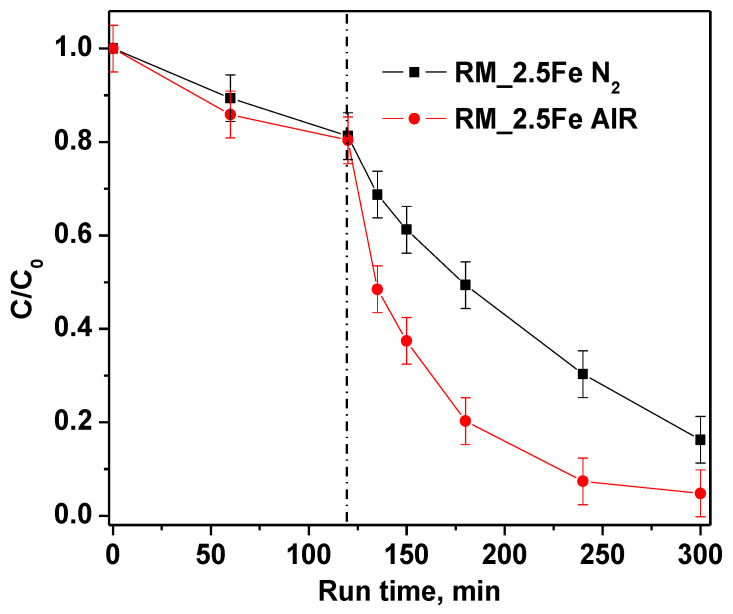
Effect of air and nitrogen atmosphere on photocatalytic discoloration of crystal violet using RM_2.5Fe photocatalyst.

**Figure 13 nanomaterials-13-00270-f013:**
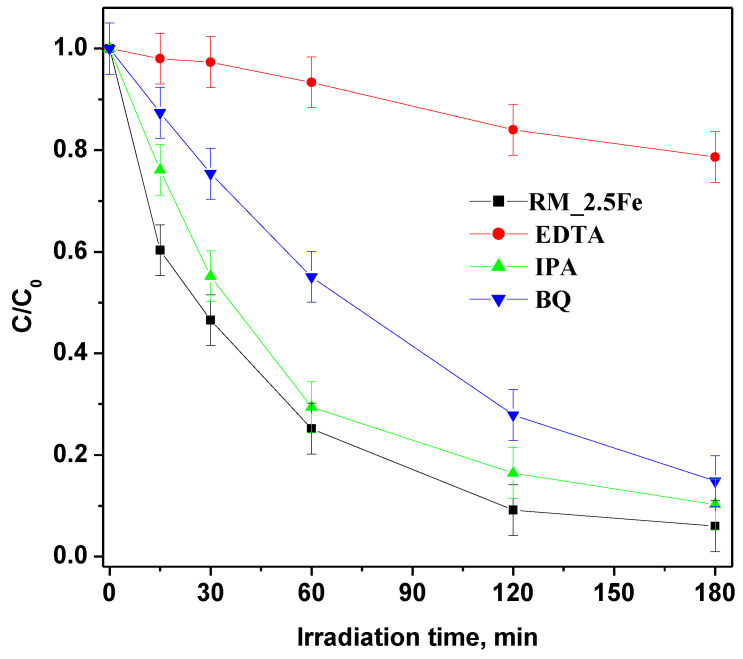
Effect of different scavenger molecules on the decolorization of CV aqueous solution using the RM_2.5Fe photocatalyst under visible (white)—light irradiation.

**Figure 14 nanomaterials-13-00270-f014:**
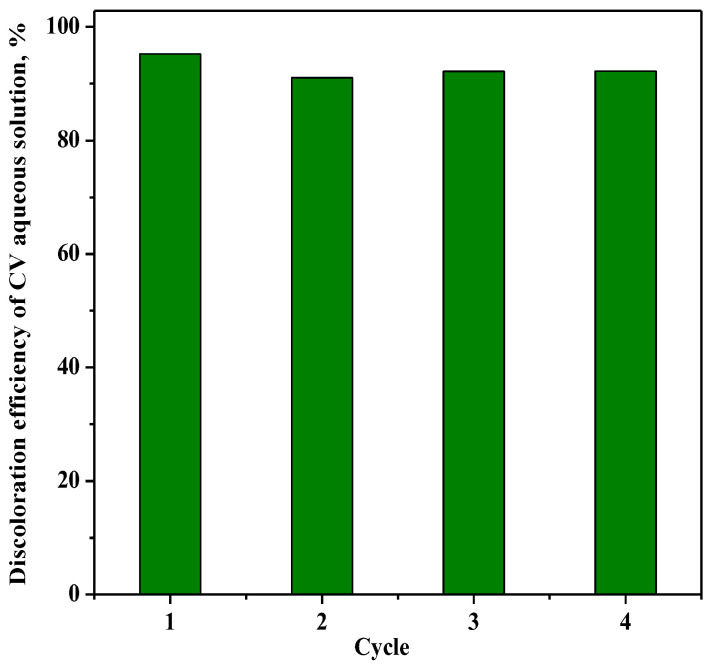
Photocatalytic discoloration of CV with the RM_2.5Fe photocatalyst under visible (white) light irradiation after 180 min: stability of the photocatalyst along four cycles.

**Figure 15 nanomaterials-13-00270-f015:**
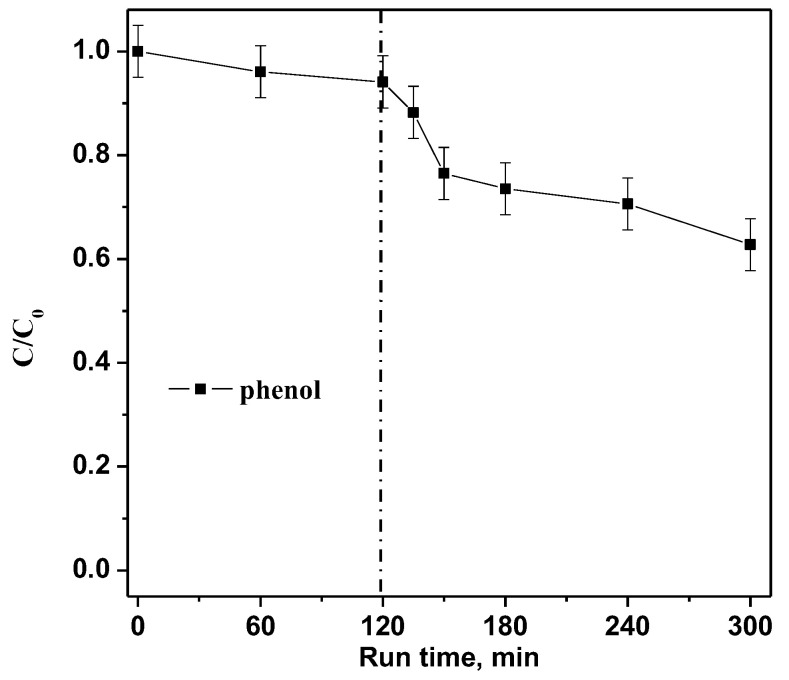
Photocatalytic discoloration of an aqueous solution containing phenol (10 mg L^−1^) with the RM_2.5Fe photocatalyst under visible (white) light irradiation.

**Table 1 nanomaterials-13-00270-t001:** QPA results concerning the abundance (wt.%) and the crystallite size (Å) of anatase, rutile and brookite in the samples.

Samples	Anatase		Rutile		Brookite	
	Abundance (wt.%)	Crystallite Size (Å)	Abundance (wt.%)	Crystallite Size (Å)	Abundance (wt.%)	Crystallite Size (Å)
RM_0Fe	84.3	74 (2)	0.1	N/A	15.6	72 (3)
RM_1Fe	82.2	82 (2)	2.5	121 (3)	15.4	73 (2)
RM_2.5Fe	54.1	71 (2)	3.6	103 (4)	42.4	75 (1)
RM_3.5Fe	51.9	65 (2)	6.6	90 (2)	41.5	79 (1)

**Table 3 nanomaterials-13-00270-t003:** TOC Removal (%) after 180 min of light irradiation.

Samples	%TOC Removal White LED	%TOC Removal Blue LED	%TOC Removal Green LED	%TOC Removal N_2_ Atmosphere
RM_0Fe	0	-	-	-
RM_1Fe	14	-	-	-
RM_2.5Fe	52	34	47	35
RM_3.5Fe	17	-	-	-

## Data Availability

Not applicable.

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
