# Peer review of "Photocatalytic Degradation of Crystal Violet Dye under Visible Light by Fe-Doped TiO2 Prepared by Reverse-Micelle Sol–Gel Method"

_nanomaterials, 2023, doi:10.3390/nano13020270_

Round 1

Reviewer 1 Report

This manuscript reported Fe-doped TiO2 devoted to crystal violet dye photodegradation under visible light by a reverse-micelle sol-gel method. The photocatalysts were characterized through XRPD, QPA by Rietveld refinement, UV−vis DRS, N2 adsorption/desorption isotherms at −196 â—¦C, ζ-potential, and XPS. All the Fe-doped TiO2 photocatalysts were active in the degradation and mineralization of the target dye, showing a TOC removal higher than the undoped sample. However, there are still some problems should be addressed before publication. Thus, I recommend the work to be published in this journal after "major revised".

Below are my concerns:

1. Reverse-micelle (RM) sol-gel method is the key novelty of this manuscript. Thus, I wonder to know what is the advantage of reverse-micelle (RM) sol-gel method. The comparison between Fe doped TiO2 by the common method and reverse-micelle (RM) sol-gel method should be added.

2. The novelty of your manuscript should be more clear in the introduction. Why did you choose Fe doped TiO2? What is the advantage and challenge of Fe doped TiO2? Why did you choose reverse-micelle (RM) sol-gel method? Some recent paper can be useful to you, such as Adv. Funct. Mater. 2021, 2108977, Applied Catalysis B: Environmental, 2023, 322, 122089.

3. Figure 6. Photocatalytic discoloration of CV aqueous solution with all the studied photocatalysts under visible irradiation (white light). RM_0Fe is better than the others? Why did you prepared Fe doped TiO2? What is the meaning?

4. The DR UV-Vis spectra of RM_0Fe(black curve) showed an onset at about 410 nm. Why did it own high efficiency for the degradation under visible light irradiation (Fig. 6)?

5. The photocatalytic and degradation mechanism need to be improved.

6. The English of the manuscript should be polished carefully when you revise your manuscript.

Reviewer 2 Report

In this work, the authors successfully synthesized Fe-doped TiO2 through a reverse-micelle sol-gel method. And the photocatalytic activity of the Fe-doped TiO2 was further discussed. However, this work lacks sufficient experimental evidence, and some key factors of the photocatalyst are not clearly defined. As a result, we cannot recommend this manuscript for publication in its current form.

The following comments should be taken for more consideration:

1.     What are the advantages of the synthesis method of this work? Why was iron chosen as the doping element? In our opinion, the authors should provide additional experimental evidence to illustrate these two issues.

2.     The author believes that there are additional Fe-oxo-hydroxo clusters on the surface of the photocatalyst. We suggest that the author make additional experiments to illustrate this view. For example, cleaning photocatalysts with sulfuric acid.

3.     The authors should describe the valence states of Fe on the Fe-doped TiO2 surface before and after the photocatalytic reaction according to XPS

4.     The SEM or TEM images of photocatalyst should be added to illustrate the micromorphology of the photocatalyst.

5.     The authors should add some basic experiments to illustrate the separation of photogenerated carriers in photocatalysts.

6.     In order to illustrate the service life of the photocatalyst, the authors should supplement the necessary recycling experiments.

Reviewer 3 Report

Reviewer’s remark: In this work, Mancuso et al. developed Fe doped TiO2 nanomaterials photocatalysts for degradation of crystal violet dye under visible light. However, there are major experimental characterizations results that needs to be provided through a a major revision addressing the comments given below, after which it can be reevaluated for publication in Nanomaterials.

Comments:

1.      What is the advantage of using the reverse micelle sol-gel method for the preparation of Fe-doped TiO2? Can't it be replaced with more straightforward reactions like hydrothermal or mechanochemial ball milling methods as reported previously? viz. https://doi.org/10.1155/2017/2191659, 10.1166/jnn.2009.c095,https://doi.org/10.1039/C2JM30360D,https://doi.org/10.1016/j.materresbull.2009.08.020etc.

2.      What is the shape of the Fe-doped TiO2 nanoparticles synthesized in this study?  Are they nanorods, nanosheets, or any other? The authors did not any information on this.

3.      FE-SEM and TEM images of the photocatalysts should also be provided to know their surface morphology and internal structure.

4.      Although the authors have mentioned that the XRD profiles were analyzed with Rietveld refinement to quantify the different phases. However, the XRD patterns should also be matched with the stick pattern of the JCPDS reference data.

5.      The author should provide the XPS survey spectrum of RM_0Fe, RM_1Fe, PM_2.5Fe, RM_3.5Fe and the comparison of Ti 2p and Fe 2p XPS core spectra to discuss the role of Fe doping with respect to the different phases of TiO2.

6.      In Tauc’s plots Figure 4b, the extrapolation lines used for the determination of band gaps should be shown in the Figure itself so that readers can see clearly.

7.      The mechanism for the photocatalytic dye degradation of CV in the presence of EDTA represented schematically should also be provided for better understanding and easy reference to readers.

8.      Is it possible to reuse the photocatalysts? If yes, how many cycles it cane be used and what is the efficiency in the CV discoloration?

9.      In general, photocatalytic activity also varies with the loading amount of the photocatalysts. The author should explore the effect of different loading of the photocatalyst for CV discoloration and its mineralization.

10.  The resolution of Figure 1 is quite low. It should be replaced with a higher-resolution image.

11.  The authors should include a digital photographic image of the photocatalytic cell used with the optimized sample before and after irradiation with white light to see the change in discoloration of CV.

12. TEM images of the photocatalysts should also be provided after the photocatalytic measurement to observe the structural stability and the change in the internal structure of the materials.

Author Response

See uploaded file

Round 2

Reviewer 2 Report

It could be published in the current form.

Reviewer 3 Report

The authors have revised the manuscript carefully addressing all the comments. Therefore, the revised manuscript is now suitable for publication in Nanomaterials.